# The Impact of CO_2_ Laser Treatment on Kevlar^®^ KM2+ Fibres Fabric Surface Morphology and Yarn Pull-Out Resistance

**DOI:** 10.3390/polym17212931

**Published:** 2025-10-31

**Authors:** Silvija Kukle, Lyubomir Lazov, Rynno Lohmus, Ugis Briedis, Imants Adijans, Ieva Bake, Vladimir Dunchev, Erika Teirumnieka

**Affiliations:** 1Institute of Architecture and Design, Riga Technical University, LV-1658 Riga, Latvia; silvija.kukle@rtu.lv (S.K.); ieva.bake@rtu.lv (I.B.); 2Rezekne Academy, Riga Technical University, LV-1658 Riga, Latviaimants.adijans@rtu.lv (I.A.); erika.teirumnieka@rtu.lv (E.T.); 3Faculty of Science and Technology, Institute of Physics, University of Tartu, 50090 Tartu, Estonia; rynno.lohmus@ut.ee; 4Department Material Science and Mechanics of Materials, Technical University of Gabrovo, 5300 Gabrovo, Bulgaria; v.dunchev@tugab.bg

**Keywords:** high modulus fibres, ballistic fabrics, CW CO_2_ laser, Kevlar^®^ KM2+ fibres, surface roughness, confocal microscopy, yarn pull-out

## Abstract

Since direct laser surface texturing of polymers is an emerging area, considerable attention is given to this technique with the aim of forming a basis for follow-up research that could open the way for potential technological ideas and optimization in novel applications. Laser pre-processing of ballistic textiles can raise surface roughness of smooth para-aramid fibres and as a result can improve the adhesion of functional coatings applied in following processing steps, thus opening new possibilities for material performance improvement. The impact resistance of ballistic fabric depends on the ability of its yarns in contact with the projectile absorb energy locally and disperse it to adjacent yarns without undergoing severe damage or failure. In addition to the yarn deformation and fracture, yarn resistance to pull-out contributes to the dissipation of impact energy significantly. The objective of this study is to optimize Kevlar^®^ KM2+ fabric surface topographies by adjusting the continuous wave (CW) CO_2_ laser parameters in such a way that it increases the surface roughness and resistance to the yarn pull-out from the fabric without destroying the unique structure of the of Kevlar^®^ KM2+ fibres. Experimental research measured data show increase in surface roughness by 50–53% and set of laser parameter variants have been obtained that allow for an increase in KM2+ 440D woven fabric yarns pull out force from fabric in the range from 50% up to 99% compared to the untreated one.

## 1. Introduction

Soft body ballistic protection panels are primarily manufactured as multi-layered assemblies of woven fabrics or other textile structures made from high-performance fibres, such as Kevlar^®^, Twaron^®^, Dyneema^®^, and Spectra^®^. The soft ballistic package structure is designed to stop high-velocity projectiles by absorbing and dissipating their kinetic energy in accordance with the required protection level [1]. The impact resistance of the fabric depends on how effectively the yarns into contact with the projectile can absorb the energy at the point of impact and transfer it to adjacent yarns. The weave structure of the fabric used in package layers have a crucial role in determining the mechanical behaviour of the yarns. Most fabrics used in the design of personal protective equipment and other impact-resistant structures are plain weave fabrics [2].

Modelling woven fabrics is difficult due to the need investigate the response both at the scale of the entire fabric and at the scale of the yarn and fibres. The second reason that causes difficulties is the variety of deformation mechanisms available to the yarns. The in-plane deformation of woven fabrics grouped into five major modes by Parsonsa et.al [3]:*Yarn stretch*—the elongation of the high-performance yarns, including para-aramid yarns approximately are linear and rate independent.*Uncrimping*—the initially crimped fabric yarns straighten via two different modes of deformation (a) uniaxial tension parallel to a yarn direction causes crimp interchange. In result the loaded yarns straighten while the yarns of the other system become more crimped, (b) biaxial tension parallel to the yarn directions causes uncrimping in both yarn systems via transverse compression of the yarns at the cross-over points.*Relative yarn rotation (trellising)*—is related to the change in in-plane angle between both yarn systems. Frequently named “shear” trellising has both an elastic component, due to rotation between cross-over points, and a dissipative component, due to the frictional resistance to rotation at these points. It is the primary mode by which large deformations are accommodated in woven fabrics.*Locking*—the resistance to trellising increases as yarns of opposite system shear-lock against one another in the plane of the fabric. When crimp interchange cross-locking occurs causing lateral contraction of the fabric to an extent that the loaded yarns jam against one another.*Yarn slip*—occurs as the relative displacement of one yarn system with respect to the other at the cross-over points. Yarn slips permanently alters the mesostructure of the fabric and occurs with little resistance under impact.

Friction between yarns plays a crucial role in how fabrics respond to impact, influencing various deformation mechanisms both directly and indirectly. Direct energy dissipation increases as the yarns within the fabric begin to move relative to each other–through mechanisms such as sliding, stretching or reorienting. The indirect effect depends on how the loads are transferred and redistributed among neighbouring yarns [4]. Although friction preserves the weave pattern and thus enhances ballistic performance, its effectiveness depends largely on the yarns’ material properties. The results show that fabrics made from yarns with higher stiffness and strength have a greater impact ballistic performance [5,6,7].

Yarn pull-out is a major mechanism for dissipating impact energy: during non-penetrating velocities, the combined processes of yarn uncrimping and translation absorb most of the ballistic energy [6,8,9,10]. When a yarn is withdrawn, yarn tension will rapidly reach a peak value named as the junction rupture force (JRF) or peak load point. During this process, the pulled-out yarn straightens as they lose crimp, causing displacement of the yarns their crossed over along their path. Thus, the JRF quantifies the yarn’s static frictional force, also known as gripping force. The imposition of greater forces produces progressive yarn slide, and the associated force become discrete. This stage is described as stick-slip type or yarn translation [8]. Yarn pull-out is influenced by various parameters including inter-yarn friction, pull-out distance, pull-out velocity, transverse pretension, fabric architecture, waviness and density, yarns linear density (tex) and modulus [4,8,11].

Several studies have investigated the yarn pull-out behaviour of Kevlar^®^ under quasi-static (QS) to intermediate loading rates, typically ranging from 50 mm/min to 500 mm/min [4,8,11,12,13]. In contrast, the novel experimental method developed by Thomas et al. [14] enables yarn pull-out testing under dynamic conditions, extending the analysis to higher strain rates. It showed that in the transition from QS to a dynamic one, increases are seen in such standard metrics for pull-out performance as peak force, displacement at peak force and energy absorbed over a specified pull distance. Beyond peak force, some differences arise during yarn translation as well.

JRF could be increased through the structure modification of fabric. The simplest way to increase the yarn wrapping angle is by weaving the fabric more tightly [15]. However, in ballistic applications, fabrics with a high degree of yarn crimp generally exhibit reduced ballistic performance [16].

Finite element (FE) modelling results indicate that the stress on the high crimp fabric fluctuated more severely at the crossover and fail earlier than that of the fabric with low crimp. In a multi-layer panel, yarn crimp continues to play a significant role in the energy absorption of each fabric layer, as well as in determining the back face signature (BFS) and the overall perforation ratio of the protective panel. Adding low-crimp fabrics to the panel increases the energy absorption efficiency of each layer compared to a panel made solely of high-crimp fabrics [17]. This is because fabric with a large crimp is prone to form high stress at the edge of the contact of the fabric with the projectile. This leads to earlier failures of the primary yarns and accordingly the energy absorbed by such a fabric is reduced due to the short engaging time. In addition, the longitudinal stress wave velocity is higher with lower levels of yarn crimp in the fabric [6,18].

Chemical treatments of plain-woven fabric have been extensively studied to increase JRF, among which resent studies witnessed the application of *shear thickening fluid* (STF) to high-performance woven fabrics for the improvement of ballistic and stab resistance. One reason is that treated woven fabrics are more sensitive to the pull-out rate than neat fabric [19,20,21,22,23]. For ballistic impact applications, moderate interfacial properties are desirable to enhance energy absorption by mechanisms like friction, slippage, cracks, and matrix debonding. Incorporating nanoparticles such as nano-silica [24], graphene [25], graphene oxide [26,27,28] has been shown to produce beneficial effects.

Zinc oxide, nano-clay, and other ceramic nanoparticles on the surface of high-performance fibres helps to overcome the limitations [29]. It has been found that the yarns treated by sub-micro-sized TiO_2_/ZnO hydrosol exhibit a nearly 50% increase in coefficient of friction compared to neat yarn, while the increase of nano-micro-sized coated yarns resulted only 10%. The authors attributed this increase to the irregular coating and rougher surface of the samples [30].

In plasma processing [31], a low-pressure plasma enhanced chemical vapour deposition (PCVD) technique was employed to treat Kevlar^®^ fabric with two different plasma types: a non-polymerizing reactive N_2_ plasma and a polymerising (CH_3_)_2_Cl_2_Si plasma. Low-temperature, non-polymerising reactive plasma etches and ablates the fibre surface, creating micro-roughness that increases fibre-to-fibre adhesion [32,33]. Microwave treatment produces comparable surface roughening and adhesion gains [34].

Several laser treatments have been successfully exploited in the textile industry in recent years. Some of treatments to obtain intended functions involve the complete or partial destruction of the material in the processing area (marking, engraving, cutting, welding, laser-based denim fading, and laser ablation) or additive manufacturing (sintering) [35,36,37,38]. Resent non-destructive techniques are gaining wider use, including fabric *fault detection* and *objective evaluation of seam pucker* [35,36] along with three-dimensional scanning for custom-fit garments production [39] and design practice [40].

The absorption of laser energy by textiles can lead to effects such as melting, plasma formation and vaporisation. These phenomena form the basis for laser-based materials processing. The localized high temperature induced by CO_2_ laser can be used to selectively convert some carbon precursors into graphene.

The laser-induced graphitization comprises various photo thermal and photochemical reactions, or thermal accumulation effect from high repetition rate laser (Figure 1).

Recent research reports confirmed that localized high temperature induced by CO_2_ laser could selectively convert some carbon precursors into graphene materials and showed that graphene could be obtained in direct scribing with laser on polyimide (PI), wood, or paper [42,43,44,45]. This breakthrough marked the beginning of a rapidly evolving field–the laser writing for producing graphene on polymer textiles. This technique allows for the straightforward fabrication of graphene-based textile electronics. Development of more practical solutions leads in parallel to an in-depth understanding of related processes. The mechanism for laser scribing of polymers was thoroughly investigated in article [42]. It was speculated that photo thermal effects owing to the long wavelength (~10.6 μm) and relatively long pulse width (~14 μs) of the CO_2_ laser caused lattice vibrations, which led to extremely high localized temperatures (˃2500 °C). PI C–O, C=O and N–C bonds could break under such high temperatures and atoms recombination occur to release the gases resulting dramatically decreased oxygen and nitrogen contents in the polymer. When the laser power was increased above 4.2 W, decomposition started to play an increasingly deleterious role in the film quality. Like the conversion of PI, the C=O and N−C bonds in Kevlar^®^ started to brake and the reorganisation of remaining carbon atoms into graphene has been ascribed to photo thermal effect induced by the laser irradiation [46,47].

Laser surface texturing (LST) is a widespread surface modification technology that involves altering the surface properties of a material by modifying its texture and roughness with high-intensity laser beams. The laser–material interaction may include localized melting, vaporisation, or ablation [48]. LST utilizes a focused laser beam to selectively remove material from the surface through laser ablation, removing layers with micrometre precision with a perfect repeatability, if applicable. Micro/nano patterning increases the surface area and aspect ratio endowing the treated surface previously uncharacteristic functional properties. It is possible to pattern a variety of materials including ceramics, metals, and polymers. Several parameters such as laser type, wavelength, pulse duration, pulse repetition rate, and laser power affect the outcome [49].

Laser patterning technique is suitable to prepare the surface creating large anchoring area of sensitive materials to obtain high-quality functional coatings. Optimised surface topographies can be elaborated according to the certain material as well as the application. Experimentally was shown the adhesive bond strength is linearly proportional to the contact area [50].

Although research is ongoing to increase JRF for ballistic performance optimization, few of them have been commercialised. The application of fabric impregnation with shear thickening fluid reduces the amount of fabric used in a ballistic, but it does not essentially decrease the overall weight due to the presence of the STF. Other chemical treatments might avoid the problem but make the fabric less flexible. Laser-based treatments can enhance yarn-to-yarn friction without increasing the fabric’s stiffness or weight [10]. Refining LST techniques for the pre-processing of ballistic textiles could raise fibre-to-fibre and yarn-to-yarn friction thereby improving the adhesion of functional coatings to the fibres later during following steps of functional coating deposition as well in composites production, thus opening new possibilities to improve ballistic performance of modified fabric or composite.

## 2. Materials and Methods

### 2.1. The Specific of Kevlar^®^ Fibre Architecture

By subjecting the aramid textile to laser induction, the surface morphology is considerably modified, while being dependent on the laser output power during the process. These resulting morphological changes can considerably increase the aramid fibre surface roughness promoting a stronger interlocking between the fibres and matrix in following applications. At the same time, it is necessary protect the fibre from the massive ablation and considerable thermal damage, which would diminish its performance. Roenbeck et al. have clearly shown the importance of characterising fibre internal structures, which reveal critical footprints of their processing conditions and, at the same time, yield essential insights into predicting fibre physical and mechanical properties. Thus, this approach offers a robust means of evaluating the factors that determine fibre performance [51]. Previous studies have shown Kevlar^®^ fibres formed by poly (p-phenylene terephthalamide) (PPTA) molecules [52]. The hierarchical structure of Kevlar^®^ fibre can be viewed at three distinct scales: the molecular structure (nanoscale), the microfibril structure (microscale) and the single fibre level (macroscale) [53]. Strong covalent bonds align the polymer chains along their axis, producing highly oriented macromolecules (Figure 2a,b). These aligned molecules form the fibrillary structure of aramid fibres, which underlies their exceptional mechanical properties [54]. Transverse to the polymer chains covalent bonding does not occur. Having stiff para linked aromatic rings and hydrogen bond donors and acceptors arranged throughout their backbone’s [55] macromolecules are linked together with hydrogen bonds and/or van der Waals forces (Figure 2a).

The inherent molecular rigidity, combined with strong intermolecular hydrogen bonding, enables PPTA molecules to achieve excellent mutual alignment (Figure 2a), resulting in a highly anisotropic unit cell (Figure 2b) consisting of covalent bonds, hydrogen bonds, and van der Waals interactions along each fundamental axis [54,55,56,57,58]. There are two types of interfaces between PPTA crystals: a longitudinal interface and a transverse interface. The longitudinal interface stands for the planes where the PPTA chain ends concentrates (Figure 3), and the transverse interface refers to the boundary between PPTA crystals that mainly includes H-bond from adjacent PPTA chains on the boundary [53].

Experiments have revealed that sets of molecules form a para-crystalline structure. The structure of highly oriented liquid crystalline polymers can be characterised by a hierarchical fibrillary structural model. Such models were first developed for the lyotropic aramid fibres in the late 1970s [56]. The study has refined our understanding of the fibrillar hierarchy and determined that the macromolecules have an average size of about 200 nm [59,60]. In the created fibre model sets of molecules adopt a para-crystalline structures [53,55,56], with crystal dimensions of 80 nm axial and ~5 nm wide [55,60], integrated into circa 22 nm wide tape-like structures [51,59], that possess axial crystallites exhibited a uniform diameter 60 nm with longitudinal dimensions of 200–250 nm [60]. Nanofibrils width in a range 15–25 nm is the main characteristic that may affect the stiffness of the core region of Kevlar^®^ fibre [51,53].

Morgan et al. found that the fibre core contain periodic transverse weak planes, roughly every 200 nm along the fibre axis—the spacing equivalent to the average length of a PPTA macromolecule (Figure 3a). They proposed that these planes are mechanically weak because macromolecular chain ends clustering close to them (Figure 3b). The absence of such transverse weak planes in the fibre’s skin indicates the presence of more random PPTA molecules chain-ends distribution [56,60].

Superimposed on the micro fibrillary structure is a pleating of the nanofibrils. Supramolecular structure investigations by Dobb et al. found out the arrangement as a hydrogen-bonded ‘pleated sheet’ morphology [55,56]. Pleats are composed from 200–250 nm long and 60 nm wide crystalline domains—microfibrils (Figure 3a). Molecular dynamics (MD) method simulation of a molecular cluster with two PPTA crystals interacting with each other clarified that the axial interaction between different pleats mainly based of H-bond interaction between PPTA chains. The stiffness and strength of each elementary pleat strictly follow the stiffness and strength of a PPTA crystal [53].

Fibre body at microscale level formed by radially clustered pleated microfibrils sheets named “ordered lamellas” (Figure 4b) [59]. Nanoscale fibre constituent’s pleats slightly offset from the fibre axis (Figure 4c) and stacked together in the direction of the fibre axis forming fibrils (Figure 4b,c).

Microfibrils are oriented along the fibre axis, with width of about 600 nm extending several centimeters. The microfibrils shown in Figure 4b joined by tie groups of chains which link one fibril to an adjacent one. This may explain the small proportion of non-oriented crystalline polymer seen in long-exposure wide-angle x-ray photograph of PPTA fibre [60]. The surface fibrils are uniformly aligned along the axis, as evidenced by the straight axial groove that remained after a single fibril was scraped off the surface. In contrast, the fibrils in the core are imperfectly packed and ordered [60].

Dobb et al. in their studies [55] concluded that several dark-field images indicate the possibility of a fibre skin-core heterogeneity, where the core is degraded selectively, often-leaving pieces of hollow skin. In the skin region, there is no explicit interface because the PPTA molecular chains are randomly distributed in the skin region. The failure of Kevlar^®^ fibre is mainly due to the failure of the interface instead of the breaking of covalent bonds in the PPTA molecular chain [53].

Transmission electron microscope (TEM) and scanning electron microscope (SEM) images of fibre cross sections have shown that skin thicknesses typically range from 0.1 to 1 µm [51,55,56,60] with fibres possessing thicker skins exhibiting lower elastic moduli and higher tensile strengths [56]. Heat-treated fibres have noticeably thinner skins, implying that post-processing triggers a molecular-level reorientation of crystallites. This reorganisation produces a more uniform structure along the fibre, enabling the material to attain a higher elastic modulus [51]. Kevlar^®^ skin is one of the main regions where amorphous PPTA locates and there are also amorphous PPTA among the PPTA crystals in the core region of the Kevlar^®^ fibre resulting in a degree of crystallization in the range of 65~75% for most Kevlar^®^ fibres [53]. The amorphous PPTA represents a homogeneous and isotropic material properties. The Young’s modulus and strength of the amorphous PPTA material are 27.05 GPa and 1.09 GPa, respectively. Young’s modulus and fracture mechanisms of a single Kevlar^®^ fibre can be obtained according to the microstructure of the Kevlar^®^ fibre, considering the distribution of pleat thickness, pleat orientation, and skin thickness [53].

Authors believe that differences in elastic modulus, tensile strength, and ultimate strain across the four classes of Kevlar^®^ fibres (Table 1) depend on how large the dispersion in crystallite orientation within micro fibrillary domains is, and that these distinct orientations become less prevalent if fibres subjected to heat treatment [51].

Thus, the thinner skins in heat-treated fibres would indicate that post-processing induces molecular-level reorientation of crystallites, resulting in more uniform structures across the fibre enabling the K49 and KM2+ achieve a high elastic modulus.

High elastic modulus fibres Kevlar^®^ KM2+ (Table 1) with diameters around 11–12 µm are often used to produce ballistic grade yarns and textile-based layers of ballistic panels and other composite structures.

### 2.2. Materials

Fabrics Kevlar^®^ KM2+ 440 type 310L (ballistic, producer SAATI S.p.A, Milano, Italy) woven from ballistic KM2+ fibres (Table 1) 440 dtex threads were subjected to the laser processing, loomstate (without final finishes), plain weave, 14 threads per cm in warp and weft direction, fabric thickness 0.17 mm, areal density 125 g/cm^2^. Commercial Kevlar^®^ KM2 fabric used for comparative analysis with the final UV protection finishing.

### 2.3. Laser Processing

Continuous Wave (CW) CO_2_ Laser SUNTOP Model: ST-CC9060, Suzhou Suntop Laser Technology Co, Ltd., Suzhou City, China) producing a continuous beam of 10.6 µm wavelengths used (Figure 5a,b).

Max laser power 100 W. Beam diameter in focus d = 100 µm and focal length F = 63.5 mm were kept unchanged. As variables in this study used laser beam Scanning Speed (*v*), Power density (*P*) and Raster Step (*∆x*) to evaluate not only impact of each separately but also their interaction effects. The 2^k^ Factorial Design is used to study the joint effects of these variables (factors) on the response and in factor screening experiments. The increase of surface roughness (qs) against the surface roughness of the unmodified fabric and yarn pullout description parameters used for response evaluation. The variation intervals and levels of factors, as well experimental data and response surface contour plots shown in Section 3. Laser-induced damages controlled visually, by colour and confocal images of confocal microscopy. Partial (the effect of carbonization is assessed based on changes in the colour of the fibres surface) or complete decomposition (the disappearance of grouped fragments of the upper fibres) are classified as unacceptable changes.

### 2.4. Surface Morphology Investigation

Confocal laser scanning microscope OLYMPUS LEXT 3D MEASURING LASER MICROSCOPE OLS5000 model “OLS5100-EAF” (Olympus Corporation, Tokyo, Japan) applied for fabric surface images obtaining and fabric surface roughness evaluation. Focused laser light scans a surface point by point making 3D topography maps. The system measures surface heights by detecting only in focus light reflections. This methodology allows 0.1 nm vertical resolution and is suitable for the woven fabrics’ steep slopes and deep ravines. A significant advantage of the confocal microscope is the optical sectioning provided, which allows for 3D reconstruction of a sample from high-resolution stacks of images [61]. Process provides non-contact measurement, real-time 3D visualization and automated data processing. They also facilitate measurement of the entire surface without damaging the surface.

Confocal images used to identify types of damage to Kevlar^®^ fibres if they have occurred during the fibres forming, fabric modification processes, and their intensity, confirming the fibrillary (~600 nm) structure of Kevlar^®^ fibres (D~12 µm). The microstructural images produced using a LMPLFLN50XLEXT W.D. 5.0 [mm] N.A. 0.6 objective (Olympus Corporation, Tokyo, Japan) with a magnification of 1125. The roughness Ra and Rz perpendicular to the marking lines with a length of 644 µm and roughness (Rq) for the entire examined area (644 × 644) µm were measured and each average Yqs value calculated from 10 Rq measurements. Samples examined immediately after laser treatment, stored in sealed plastic bags without light access for further testing.

SEM Helios 5 UX (company Thermo Scientific, Eindhoven, Netherlands, The Netherlands used in the Institute of Solid-State Physics, University of Latvia) used to investigate Kevlar^®^ fabric samples surface morphology before treatment.

SEM ZEISS (Oberkochen, Baden-Württemberg, Germany, used in the Vasil Levski National Military University, Bulgaria) used to investigate laser processed Kevlar^®^ fabric samples surface morphology. Sputter coatings with gold were applied to prevent non-conductive Kevlar samples charging and enhance image quality.

### 2.5. Quasi-Static Yarn Pull-Out Setup and Test Methodology

An initial yarn pull-out test performed using Kevlar^®^ fabric to ensure that the experimental results are reproducible and accurate. As shown in Figure 6a, the yarn pull-out load and displacement response are generally comparable to published literature results [7,62]. The pull-out response described in three different zones. In zone 1, the pull-out force increases linearly as the pulled yarn undergoes an uncrimping and straightening process, which resisted by the static friction of the orthogonal yarns of the woven fabric. The end of zone I is characterized by the maximum force due to static friction P_static_. Total uncrimp length obtained from load–displacement curve zone I: Δ*L*~3.52 mm, where the tail length is 10 mm, but the test fabric height (L_0_) is 50 mm. The thread pull-out speed was 50 mm/s according to the method described in the literature [7,62].

When the applied tension surpasses the static-friction threshold, the yarn transitions into sliding, which marked by a sharp drop in force at the beginning of zone II. In II zone, the nature of the movement goes from static to kinetic. The oscillations in zone II are due to the sliding behaviour when the pulled yarn passes through the orthogonal threads. In zone III, produced progressive descent of the pull-out force as the effective number of cross-yarns decreases.

The length of zone II is approximately equal to the tail length l0 mm of the sample. Finally, in zone III, the force decreases to zero as the length of the yarn tail crosses the fabric and fully pulled out. The length of zone III is approximately equal to the test fabric height L_0_.

Thread-pulling experiments were performed with the device Instron Force transducer model 2519-107 (capacity: 5000 N, model 2519-107, ID:3345K6537, Illinois Tool Works Inc., Norwood, MA, USA) and results were observed with the program Instron Bluehill Lite Version 2.17., SN: 717479C, Illinois Tool Works Inc., Norwood, MA, USA). Ten threads pulled for each sample, and every fifth thread pulled out.

Two samples have been prepared for each variant in a size of 220 mm in the warp and 110 mm in the weft direction. According to the methodology [7,62], warp threads were unrolled from the bottom of the sample to a width of 10 mm, and from the top to a width of approximately 60 mm, so that the height of the tested sample remained 50 mm.

The weft threads pulled vertically upwards. To ensure uniform horizontal tension of the sample, the edges of the sample (30 mm) folded and sewn with a straight stitch sewing machine. Bars inserted at the fold point along the entire height of the sample, which ensures uniform horizontal tension of the sample.

The sample secured in a self-made mounting frame (Figure 6b), the structure of which consists of two threaded rods, which, with nuts, secure one movable and the other fixed vertical part of the frame, in which the sample is mounted. Before the thread pull-out, the fabric sample tensioned in the frame with a tension of 100 N using Instron 2519-107.

### 2.6. Experiment Design

Statistical experimental designs implemented in two steps:

***I step*** to set boundaries within which changing the values of the laser parameters does not lead to the surface fibres evaporation and/or carbonization using full factorial experiment 2^3^ (Table 2 in Section 3) to see not only how each factor individually affects the response, but also how the factors interact and influence each other;

***II step*** to investigate the effects of laser power (*P*), raster speed (*v*), and raster step between lines (*Δx*) and their interactions to the Kevlar^®^ KM2+ fibres surface roughness and yarns pull-out resistance using factorial design 2^2^ and obtained the fibre surface protective laser parameter limits in I step (Table 4).

As response, parameters ***qs mean increase percentage*** to compare with the laser untreated sample Yqs and the junction rupture force (JRF) laser treated samples used.

In both steps, experiments are duplicated, every response value calculated as average of 10 primary measurements in each duplicate, in total 20 primary measurements for each set of factors. The correspondence of the calculated mathematical model in the form of incomplete higher-order polynomial to the experimental data is assessed using the F (Fisher) criterion at a 5% confidence probability. In both steps F_calc_ < 1 indicates a very good compliance. The analysis was carried out by studying the response surfaces and corresponding contour plots which placed in Section 3.3 and Section 3.4 and graphically describe the dependence of the modified fabrics roughness and JRF on the considered laser processing parameters.

All tests carried out in typical lab conditions, what is provided by specialized laboratory climate control chambers. Samples pre-cleaned with acetone for 2 min before processing, then rinsed with distilled water, dried and stored in sealed plastic bags without light access for further processing and testing.

## 3. Results

### 3.1. Laser Unprocessed Kevlar^®^ Fabric Surface Morphology

The macro-level unevenness is created by the structure of the Kevlar^®^ fabric. This type of unevenness specially pronounced in the plain weave fabrics due to frequent intersections of weft and warp yarns. It was determined by the interaction of the weave and the linear density of the threads, as well as the technological parameters applied during the weaving process. In ballistic fabrics, untwisted yarns are mainly used. Without twist to bind them, the fibres move freely: they group together during over—and—under yarn passes, cross each other and leave voids on the fabric surface. These irregularities alter the surface morphology at both macro and micro levels, making the fabric markedly uneven in all three dimensions (Figure 7).

Optional micro- and nano-scale unevenness formed by the fibres structure defects, as well as the finishes applied to the threads and fabric during manufacture processes (Figure 8 and Figure 9). Since fibres on the fabric surface are arranged at varying heights, those protruding above the surface are more likely to be damaged or subjected to harsher treatment during the fabric post processing.

Aramid fibres readily fibrillate under abrasion, especially when the force acts perpendicular to the fibre axis—the stress direction experienced by warp threads during weaving. This mechanical damage produces micro- and nano-scale debris, such as slivers of fibre skin, fractured filaments, micro and nano-fibrils. These particles can stay attached near the damage site or detach and migrate across the fabric, where they accumulate into contamination clusters on the fabric surface (Figure 8 and Figure 9).

Figure 8a reveals a tear in the fibre’s outer skin that also exposes part of the core—its inner microfibrillar structure. Figure 8b shows a detached skin fragment accompanied by a cluster of microfibrils. When the skin damage is minor, the fibre’s core architecture remains intact, as illustrated in Figure 8c.

In fibre damage locations, where there is a relatively high concentration of micro fibril clusters, local fibres destruction could occur during the treatment process even at relatively low laser power creating local carbonized clusters.

Clumped fibres and nano-level particles on the fibres surface (Figure 9Ia) may have resulted during fibres forming or from finishes applied during previous technological processes. On the right side of the image, the micro size clusters (Figure 9IIb) and UV protection coating (Figure 9IIc) on fibres can be seen, which may create unexpected effects during the laser processing (Figure 9II).

### 3.2. Laser Irradiated-Surface Structures

Due to macro-, micrro and nano scale balistic Kevlar fabric surface unevenness, as well as the finishes applied to the threads and fabric during manufacture processes the woven fabrics laser processing can be challenging if you want to preserve the inherent properties of the fabric.

Laser surface texturing (LST) can adapt material properties such that it can enhance adhesion characteristics [63], alter friction coefficients [64] and reduce the wear rate [65]. Laser surface patterning based on self-organized laser irradiated structures and direct laser-inscribed structures. Self-organized means that although the surface is irradiated using a homogeneous spatial beam, the topography of resulting surface can be described as quasi-periodic surface morphology. Laser interaction with material leads to material removal (ablation) and eventual formation of micro/nano structures. Surface micro/nano structuring creates many different structures on various materials. The two broad types are *laser-irradiated structures* and *laser-inscribed structures*. The possibilities are limitless for *laser-inscribed* structures, whereas for *laser-irradiated* structures only two main types—random and periodic structures reported. *Laser-irradiated* nanostructures in form of nanoholes, nanocavities, nano protrusions, nano bumps and nano rims can appear alone, or they can be associated with microstructures [66]. Thus, the self-organized surface structures may consist of microstructures, nanostructures, or hybrid variants (Figure 10). So far, there are few reports on the range of applications of the laser-irradiated structures compared to the wide range of applications offered by laser-inscribed structures [67,68]. The possible technological constraints of laser processing are largely determined by the expected processing effects and the structure specifics of material subjected to processing and its diverse properties as material. During the first step in a laser-induced material breakdown photoionization, valence electrons absorb enough energy to move to the conduction band where free electrons generated by multiphoton ionization or by impact ionization. In the case of infrared (IR) radiation at increased kinetic energy and material removal through melting or vaporization occurs [68]. Kevlar^®^ does not melt but decomposes at relatively high temperatures (427 °C to 482 °C) in air and approximately at 538 °C in nitrogen, when tested with a temperature rise of 10 °C/min. Decomposition temperatures vary with the rate of temperature rise and the length of exposure [69,70,71]. Thus, laser irradiated structures on Kevlar^®^ fibres surfaces can formed during the decomposition process, releasing volatile chemical groups.

CO_2_ laser irradiated random submicron structures seen on KM2+ fibres surfaces (Figure 10a). Larger sized cavities have formed with the fiber bordering a deep ditch (Figure 10b) or near fibre surface damage, for example, when the fibre skin is partly peeled off (Figure 10c).

In laser processing with lasers operating in continuous mode (as in experiments under discussion), the threshold for a specific technological process of radiation interaction is determined by two factors: power density and exposure time. These factors are calculated using the following formulas:*q_s_ =* 4*P*/d_o_^2^(1)
where *P* is the power in *W* and d_o_^2^ is the diameter of the laser working spot on the surface in mm.Δ*t* = d_o_/*v*(2)
where Δ*t* is the exposure time and v the speed in mm/s.

As d_o_ = 0.1 mm (const) and scanning speed v in a range from 100 to 150 mm/s, then experimental exposure time Δ*t* varies within the limits of 667 to 1000 µs (0.67 to 1 ms).

### 3.3. Influence of Laser Parameters Interaction on Kevlar^®^ Fabric Surface Roughness

The response of polymers to laser influenced by the nature of the polymer and parameters of the laser beam. Although the primary outcome of light absorption is free electrons or phonons and not heat, the long interaction time associated with long laser pulses produces heat from the collision of these free electrons. Heat energy transfer through the polymer matrix from absorption and collision of photons leads to rapid temperature rise within the bulk. Depending on the amount of energy input, thermal degradation may occur. Using different investigation techniques, changes in the optical or physical properties of polymers indicate whether cross-linking or degradation dominated during laser irradiation [72].

The magnitude of fluency necessary for ablation to happen is influenced by the material’s absorption mechanism, the presence of defects, the surface morphology, the microstructure, and laser parameters such as wavelength and pulse duration [73]. The optical images of experimental untreated and laser treated fabric samples seen in Figure 11. The upper fabric layer carbonization can be recorded visually for both laser-treated fabric samples (Figure 11b,c) compared to unprocessed fabric (Figure 11a). The carbonization intensity depends on fluence values correspondingly 36 and 40 J/cm^2^, although the fluence difference is small, the surface degradation of sample **c** is significantly more intense compared to sample b, seems to have too low scanning speed for both samples.

Ablation occurs through photochemical, photothermal, or mixed mechanisms, governed by the laser’s settings (pulse duration and wavelength) and the material’s optical properties (reflectivity and absorption coefficient) [74]. In the photothermal process, the energy from laser pulses increases the surface temperature of the material, leading to melting and/or vaporization. Photothermal processes induce surface modifications such as roughness in polymers. However, in the photochemical process, there is a direct breakage of molecules by highly energised photons incident on a material surface, hence inducing chemical modifications. The combination of both photothermal and photochemical processes modify the roughness and chemistry of surfaces simultaneously [75]. IR and visible laser pulses induce photothermal effects, while UV radiation able to produce photochemical effects. Micrographs in Figure 12 show that the allowable power for KM2+ fibres has been exceeded and both processes have resulted in unacceptable fibres degradation and evaporation.

Confocal and colour micrographs corresponding to the sample seen in photo image (Figure 12c) both show partial carbonization of surface fibres (white frame) or even group wise decomposition of fibre fragments into volatile compounds (yellow frame). Power 1.6 W, exposition time 0.002 s (2 ms). At the same time, early pilot experiments at lower powers showed a significant increase in surface roughness because of laser treatment. To implement the experiment outlined in step 1 according to the variation levels of the laser parameters shown in Table 2, and in result of the factorial experiment design 2^3^ carried out, a regression Equation (1) was obtained for the surface roughness measurements of the treated samples (8 variants in two replicates). Within the considered limits of factor changes (Table 2), equation describes both the intensity of each effect on the increase in surface roughness Yqs relative to the surface roughness of the unmodified fabric and allows for the evaluation of interaction effects.polymers-17-02931-t002_Table 2Table 2The 2^3^ factorial design levels.FaktorsDesignation
Coded xi Values

−101**Interval**×1*qs*, W/cm^2^5.8 × 10^3^1.1 × 10^4^1.6 × 10^4^5.1 × 10^3^×2*v*, mm/s7588100130×3*Δx*, µm407010030


Laser power directly affects the amount of energy delivered to the material. For delicate materials lower power levels are suitable as excessive power can lead to overheating, material damage, and effects like cracking or excessive evaporation. Slower scanning speeds (longer exposure times) mean the laser dwells longer on a spot, allowing more heat to be absorbed by the material. This can lead to deeper penetration, wider melting pools, or more significant changes in surface properties. Excessive exposure time can cause overheating, leading to issues like material evaporation, cracking, or undesirable changes in microstructure. As can be seen in the micrographs of Figure 13, different combinations of the levels of the three factors are essential, resulting in different surface modification effects.

Comparing images of Figure 13a1,c, one can confirm that the fluence values are the same in both cases, but the resulting effects are very different—if the surface fibres of (Figure 13a1) have a high degree of degradation, the surface fibres of variant (Figure 13c) are virtually undamaged. On the other hand, in variant (Figure 13a1), the degree of fibre degradation is very high. It is difficult to claim that differences are caused by changes in raster steps based on images. The impact intensity has quantitatively characterized by the coefficients of Equation (3) calculated from experiment data:(3)$$\mathrm{Yqs}=38.4\%-{1.6\%x}_{1}-{1.8\%x}_{2}\;+\;{2.2\%x}_{3}-{3.9\%x}_{1}x_{2}-{2.0\%x}_{1}x_{3}-{1.6\%x}_{2}x_{3}-{7.1\%x}_{1}x_{2}x_{3},$$
where Yqs surface roughness increase in relation to the raw sample, %, *x*_1_*, x*_2_*, x*_3_ coded values of power density *P*, raster speed *v* and raster step *Δx* respectively.

The coefficients of the Equation (3) suggest effects of mutual factor interaction more important than the factors *P*, *v* and *Δx* separately in their range under study. Graphical interpretation of the equation by corresponding response surface contour plots fixing *Δx* at levels 40 and 100 µm respectively seen in Figure 14.

The greatest increase in Yqs (50–53%) achieved if *Δx* = 100 µm combined within the considered limits with the higher power range (1.18–1.26) W and lower raster speed *v* range (75–77) mm/s (area marked blue green in a white rectangle, the upper left corner). Relatively slightly smaller Yqs increase (47–50%) seen if *Δx* = 100 combined *lower power range* (0.46–0.58) W and *v* in a range (94–100) mm/s (area marked brown in a white rectangle, lower corner on the right) (Figure 14). Based on the analysis of the results of the experiment carried out in step 1, the limits of changes in the laser parameters planned for the experiment in step 2 determined, and the raster step fixed as seen in Table 3.

### 3.4. Quasi-Static Yarn Pull-Out

In ballistic protection, the yarn pull-out plays a significant role in woven fabric-based soft body armour. The front layers of soft BP provide the buffering effect, and the velocity of the projectile is reduced. When the bullet engages with layers away from the impact face, yarn pull-out is still prominent in energy absorption.

Thread-pulling experiments performed to assess the laser processed effects on the modified Kevlar^®^ fabric determining the maximum force with which a thread is pulled out of the fabric.

The first layers of the fabric perceive the first bullet impact, so it is important where the bullet hits—at the intersection of the threads or between them. Nilakantan et al. [76] studies of ballistic fabric penetration during a ballistic event discovered that yarn pull-out plays a dominant role in energy dissipation if the bullet strikes in the crossover of yarns. That increases the possibility of yarn pull-out, and therefore fabric penetration is less likely to occur. If the bullet strikes between the yarns, the yarn pull-out effect is reduced, and hence higher is probabilistic fabric penetration by projectile.

Two methods are used in thread-pulling experiments—pulling the thread from the middle of a horizontally fixed fabric (centre pull-out) and pulling the thread from a vertically fixed fabric. In thread pulling from the middle, a loop of thread is pulled from both sides of a horizontally fixed fabric. Kirkwood et al. [8], have studied this where the contribution of yarn pull-out to energy absorption will be quantified by measuring the extent of yarn pull-out in the impacted fabrics and then utilizing the quasi-static yarn pull-out mode. They concluded that projectile defeat can be achieved without yarn fracture, and yarn uncrimping and yarn translation are significant mechanisms of energy absorption. Yarn translation is most prevalent for targets with fewer fabric layers, with yarn uncrimping becoming more significant as the number of fabric layers’ increases. In result of study authors concluded that the ability to quantitatively verify the accuracy of this approach is limited by the inability to directly observe yarn uncrimping in impacted targets, restricting measurements to yarn translation only.

In recent years, scientists have increasingly used the thread-pulling method from a vertically fixed fabric [77,78,79]. The frictional slip between warp and weft threads can be characterized by a yarn pull-out test, which is usually recorded using a tensile machine. The fabric responses are described as follows: when the yarn is pulled out, the yarn tension will quickly reach a maximum value, the junction rupture force (JRF) or peak load point [80,81]. During this process, the pulled yarn becomes uncompressed, causing displacement at the intersections in its path. In this regard, the JRF or maximum load point is considered as a measure of the static friction force or gripping force of the yarn. The application of larger forces results in progressive yarn slip, and the associated force becomes discrete. This stage is then characterized as slip mode or yarn translation [4]. The setup used for the yarn pull-out test in this research designed to clamp the side edges, and the yarn tails. When the bottom edge is clamped, the yarn itself must be left unclamped; otherwise, the force transmitted to the load cell is the yarn tension and not the pull-out force.

As force is applied, fabric displacement and crimp extension occur (Figure 15) before the static friction is overcome (i.e., up to peak force). During the stage 1, yarn crimp extension (uncrimping) occurs by yarn progressively straighten. In the yarn translation region, the stick–slip phenomenon take place (stage 2) as the pulled yarn passes over the orthogonal yarns while being pulled out of the fabric. Forces tend to oscillate between local maxima and local minima. While stick–slip corresponds to force drop from maximum to minimum (i.e., pull yarn being released from the fabric), the force increase from minimum to maximum corresponds to accumulative retraction force (stage 3) (i.e., response of remaining pulled yarn that is not yet released from the fabric) [81].

Considering the speed change range used in the previous experiment insufficient, the range moved to higher speed range by setting the scanning speed lower limit at 100 mm/s without changing the power density range (Table 3).

Results of pullout tests of laser-processed samples modified according to the laser processing parameters fixing raster step *∆x* = 80, shown in Table 4 and corresponding contour plots of JRF in Figure 16.polymers-17-02931-t004_Table 4Table 4Evaluation of laser power density and raster speed on yarn pullout resistance from the fabric structure.
Junction Rupture Force, NTensile Stress,%Crimp Extension, mmMean JRF Increase *
*Range**Mean**±**Range**Mean**±**Range**Mean**±*
**KM2+440**0.382.420.041.22.372.151.52.860.16
**V1_laz**0.533.650.050.72.102.150.82.630.0950.8%**V2_laz**0.804.820.081.22.742.151.53.300.1499.4%**V3_laz**0.424.680.051.22.742.151.53.300.1593.5%**V4_laz**0.584.10.070.82.161.021.02.180.1369.7%* Increase regarding JRF of unmodified KM2+ 440. Since the JRF corresponding relative standard errors do not exceed 1.7%, the error of the mean JRF increase was not calculated.


In result of laser processing, without destroying the Kevlar^®^ KM2+ fibres 440 dtex yarn pull-out max strength JRF increased by 50 to 99% compared to unmodified fabric (Table 4). Areas with the higher JRF values in a range from 4.4 to 4.9 N marked with a white dushes rectangle in the upper left and lower right corner of Figure 16.

## 4. Discussion

Results of thermogravimetric (TG) analysis coupled with the Fourier transform infrared spectroscopy showed that the decomposition of Kevlar fibre has experienced three stages: immediate free water release from fibres at the first stage (100 to 240 °C), after that the weight reduction curve remains relatively flat until decomposition starts; the dehydration and partial depolymerization which shorten polymer chains happen at the second stage (240 to 420 °C); at the third stage fibres fragments further react and produced the gases of small molecular mass, and the main products are water, ammonia, carbon monoxide and carbon dioxide (427 to 482 °C in air and about 538 °C in nitrogen). The weight loss of Kevlar fibre is quite slow before decomposition temperature. The third is the main stage of the decomposition, and the residues finally reach 56 mas. % if heated in the air [70,82]. Although in the experiment laser processing in air was carried out with relatively low power values ranging from 0.45 to 1.26 W, nevertheless, the local temperatures can reach the decomposition threshold in result of relatively prolonged exposure in the range of 0.67 to 1 ms, causing partial decomposition of the surface layer of the fibres. The gases of small molecular mass released during the process possibly cause the changes in surface topography observed in SEM micrographs (Figure 10, Figure 17 and Figure 18).

In the case of infrared (IR) radiation, photoionization is the first step in a laser-induced material breakdown. During this step, valence electrons absorb enough energy from incident laser photons to move to the conduction band. In this band, free electrons are generated by multiphoton ionization or by impact ionization. At increased kinetic energies, subsequent collisions of energized free electrons result in the emission of secondary electrons through inverse bremsstrahlung absorption. This leads to an avalanche growth in the number of free electrons and, finally, material removal through melting or vaporization in case of Kevlar [83,84].

Developed model for predicting the contribution of different structuring mechanisms (ablation and swelling) in the direct laser structuring of polymer substrate [85]. Assuming that the swelling process can be linked to the absorption of IR photons by the fibres surface, which then formed gaseous by-products, creating pores (Figure 17c,e) and resulting in a localized volume increase in form of nanoscale bump (Figure 17a). Such can also be seen around the peeled skin fragment (Figure 17b) and microscale structures (Figure 10a and Figure 17d), which has already been there before laser processing.

Improvement of interface roughness contributes to the enhancement of yarn pull-out max strength JRF. Laser processing enhances these mechanical properties and increases surface roughness without disrupting the underlying fibre architecture (Figure 10, Figure 17 and Figure 18). Laser-irradiated micro and nanostructures randomly located on the surface of the fabric-forming fibres in form of nanoholes, nanocavities, nano protrusions, nano bumps and nano rims with different sizes.

Nano sized protrusions are dominating (Figure 10, Figure 17 and Figure 18). More severe damages associated with defects that have arisen in the processes of fibres extraction and/or fabric formation such as deformation of the fibres cross-section seen in Figure 17, torn skin (Figure 17b) or fragments of finishes used in the weaving (Figure 17d and Figure 18b). Laser fluence may affect depth and sizes of nano/micro holes and cavities (Figure 17 and Figure 18). On the other hand, to theoretically justify the impact of this various surface effects on roughness and yarn JRF, deeper research is needed.

The experimentally obtained data suggest that within the examined ranges of laser parameter variations, optimal combinations can provide both greater fluences in a range from 13 J/cm^2^ to 16 J/cm^2^ (deeper surface structures) and with less fluences in a range 4 J/cm^2^ to 7.5 J/cm^2^ (negligible sized nano protrusions dominates). In both ranges the Kevlar^®^ KM2+ fibres fabric offers the greatest resistance increase within the examined limits to the thread withdrawal compared to the raw one.

## 5. Conclusions

Laser processing within the range of used parameters can modify the upper layer of the smooth Kevlar^®^ fibres surface without causing damage to the fibres core structure, thus increasing the resistance to the extraction of threads from the fabric structure, as in ballistic protection the yarn pull-out plays a significant role in woven fabric-based soft armor performance.

It was experimentally confirmed that applying processing with the examined set of laser parameters can increase this indicator without the thin fibres skin total breakage and causing damage to the core structure of the ballistic Kevlar^®^ KM2+ fibres. Laser surface patterning based on self-organized laser irradiated structures formation on the fiber surface can increase the roughness of the fiber surface and, consequently, also the contact area in subsequent coating and composite formation processes.

It can be assumed that performing surface treatment with an IR laser in an inert atmosphere or vacuum could increase the Kevlar^®^ KM2+ decomposition threshold, which could in turn allow for higher permissible processing power and reduce exposure time.

At the same time, data was obtained based on a specific Kevlar^®^ fibers type, a fixed fabric structure designed for a particular application using IR laser and cannot be simply transferred to other situations. There are too many variables in the set, each of which can significantly influence the outcome directly or indirectly.

Future research should be focused on specific applications. Thus, by reducing the number of variables, it could increase efficiency and lead to a positive outcome more quickly or timely detect that laser processing does not provide the expected effect in achieving the set goal.

## Figures and Tables

**Figure 1 polymers-17-02931-f001:**
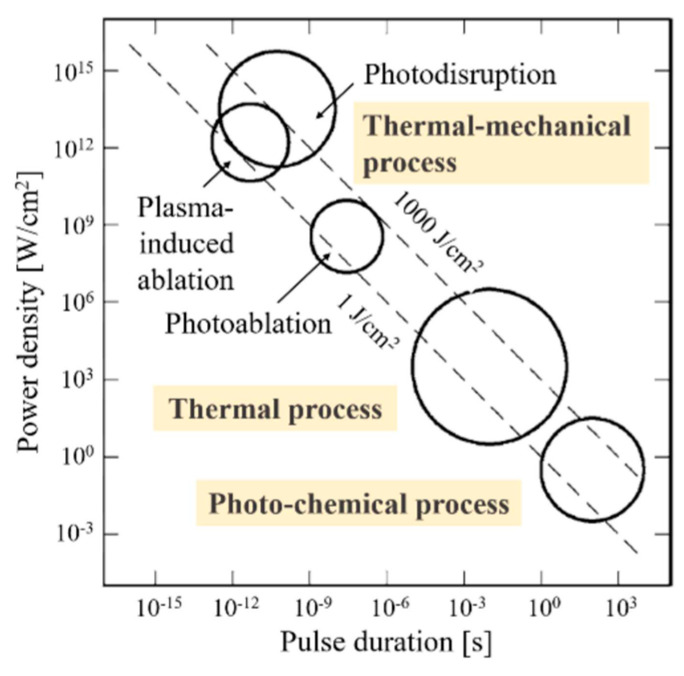
Laser–tissue interaction [41].

**Figure 2 polymers-17-02931-f002:**
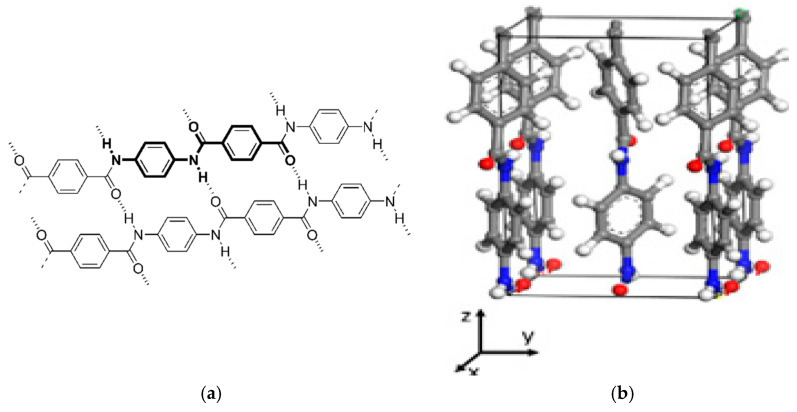
Highly oriented PPTA macromolecule chains (**a**) and crystal cell (**b**) [53].

**Figure 3 polymers-17-02931-f003:**
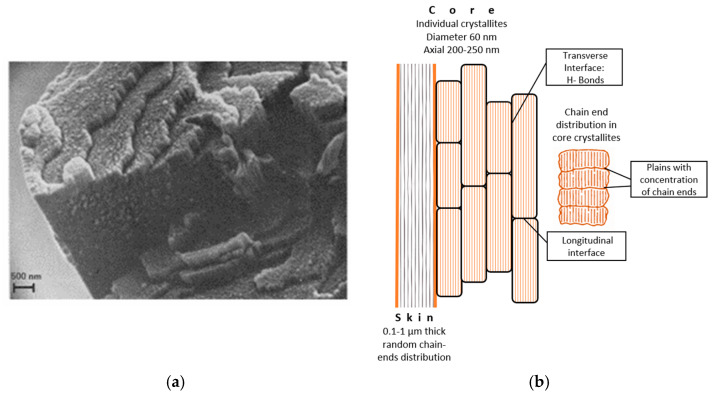
Scanning electron micrograph of the fracture topography of Kevlar^®^ 49 fibre [56] (**a**) and chain end distribution in core crystallites (**b**).

**Figure 4 polymers-17-02931-f004:**
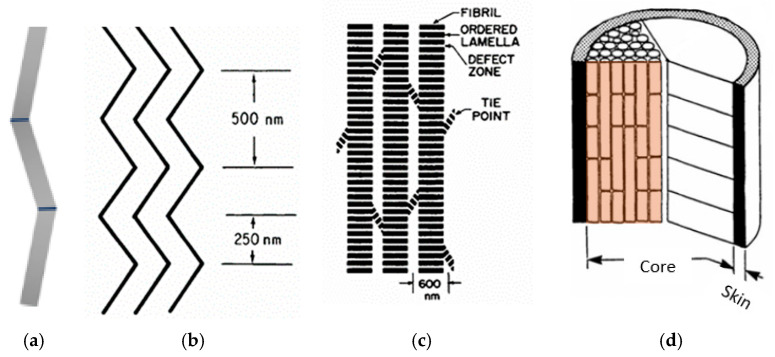
Schematic illustration of the nanofibril (**a**) [53], nanofibrils pleat (**b**) [60], microfibrils (**c**) and fibre (**d**) [60] structure.

**Figure 5 polymers-17-02931-f005:**
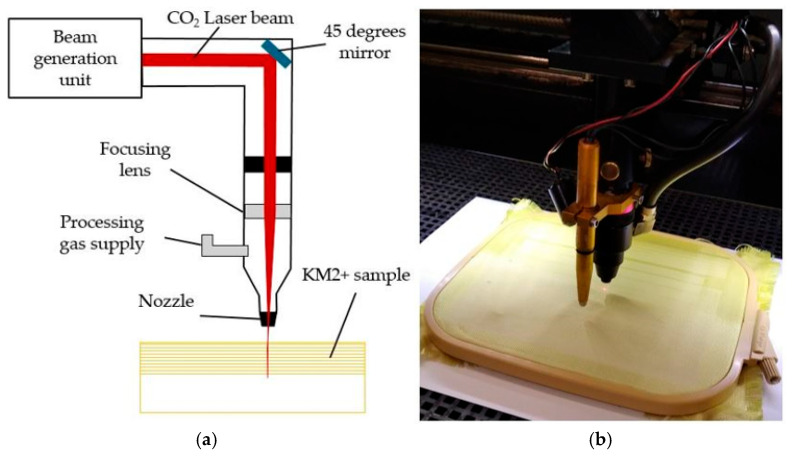
Experimental set up of laser processing.(**a**) processing principle, (**b**) laser fabric surface texturing.

**Figure 6 polymers-17-02931-f006:**
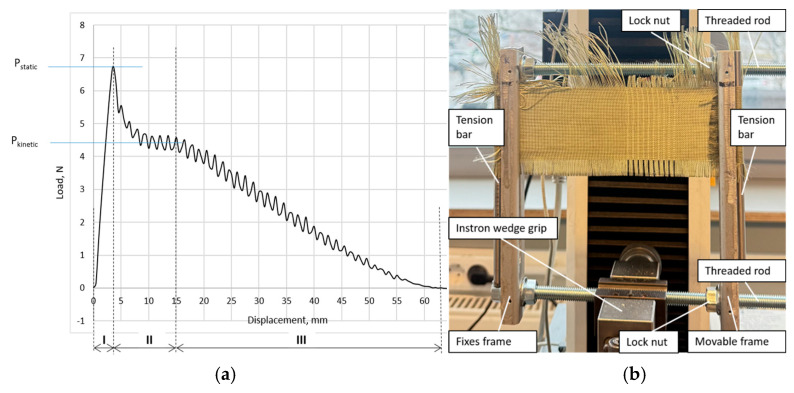
Load displacement curve from pull out test (**a**), sample mounting frame (**b**).

**Figure 7 polymers-17-02931-f007:**
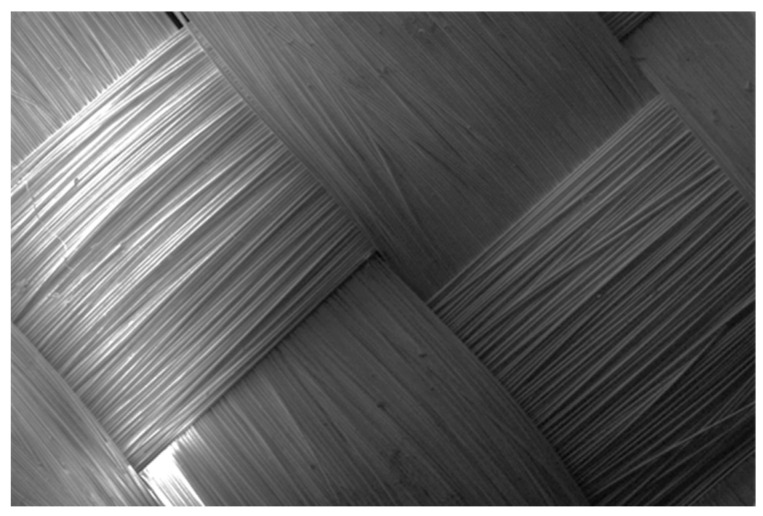
Uneven Kevlar^®^ KM2 fibres arrangement in the 600 dtex yarn over passes of plain weave fabric.

**Figure 8 polymers-17-02931-f008:**
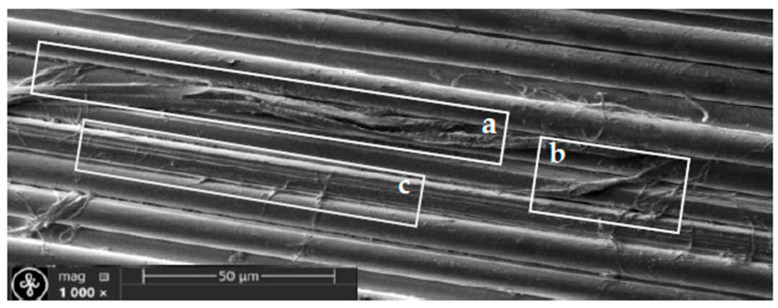
Micro- and nano scale fibres damage and leftover clusters. (**a**) damaged fiber skin and inner structure, (**b**) torn skin pieces, (**c**) minor skin damage, the internal structure has been preserved.

**Figure 9 polymers-17-02931-f009:**
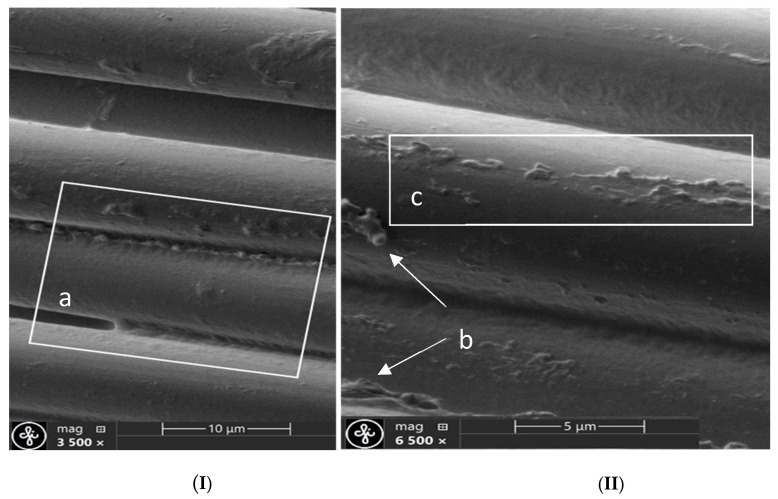
Nanoscale surface defects. (**Ia**) fibers stick together and nanosized particles, (**IIb**) difficult identifiable micro size clusters, (**IIc**) UV protective finishing.

**Figure 10 polymers-17-02931-f010:**
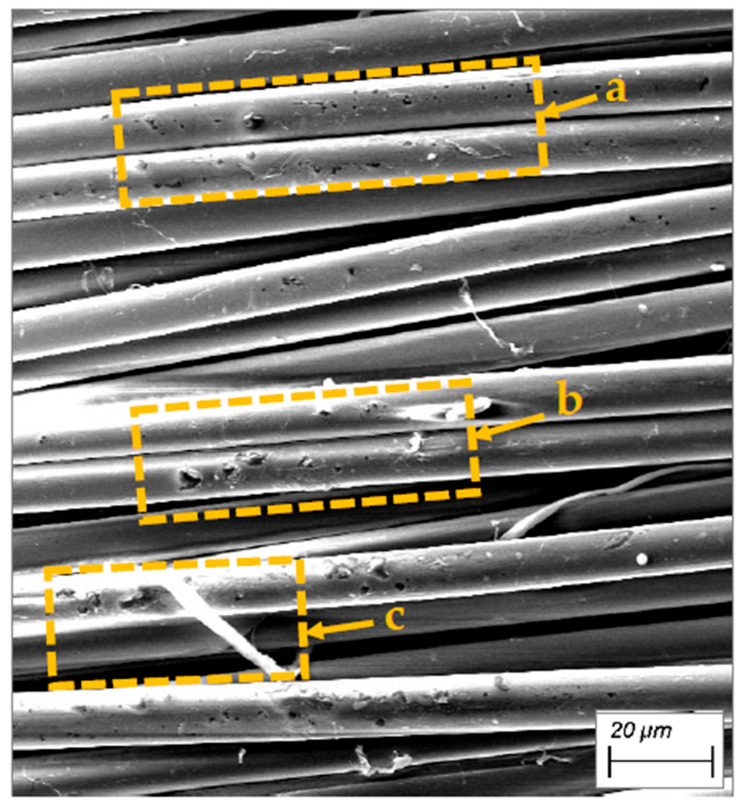
Laser-irradiated structures on KM2+ fibres. P = 1.6 × 10^4^ W/cm^2^, v = 100 mm/s; *∆x* = 80 µm. (**a**) random submicron holes on both fibres and irregularly shaped micro sized cluster on the surface, (**b**) relatively large size cavities when fibre bordering a deep ditch, (**c**) near the torn fragments of the skin.Surface modification during laser interaction with a material occurs when the target material absorbs sufficient energy from laser irradiation during exposure time, raising the energy of its atoms or molecules above their binding energy. This energy absorption leads to various structural transformations. The response of the material to laser irradiation, which governs the different mechanisms of radiation interaction, is determined by the laser exposure time, power density, and the physical properties of the material [67,68].

**Figure 11 polymers-17-02931-f011:**
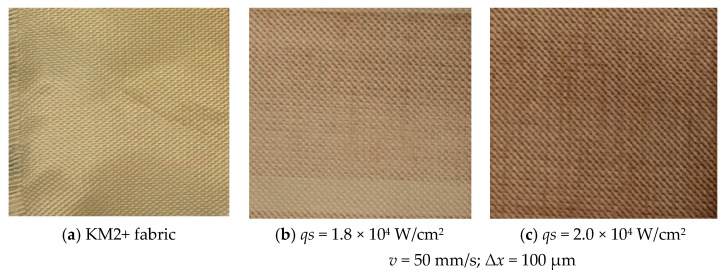
Fabric surface before (**a**) and after (**b**,**c**) laser processing.

**Figure 12 polymers-17-02931-f012:**
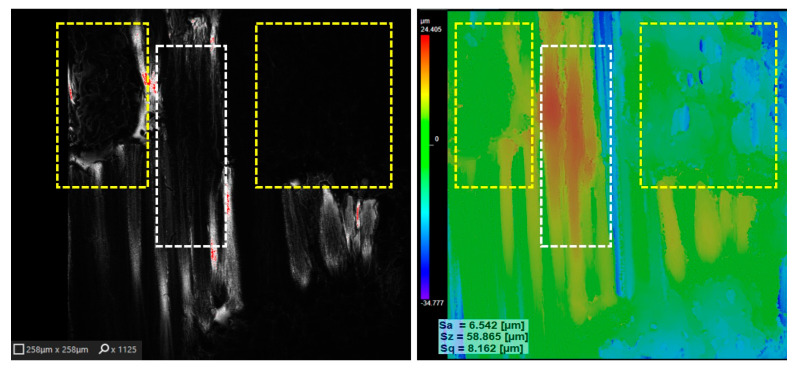
Confocal (**left**) and colour (**right**) laser processed KM2+ fibres fabric (*q_s_* = 2.0 × 10^4^ W/cm^2^, *v* = 50 mm/s; *Δx* = 100 µm) surface micrographs.

**Figure 13 polymers-17-02931-f013:**
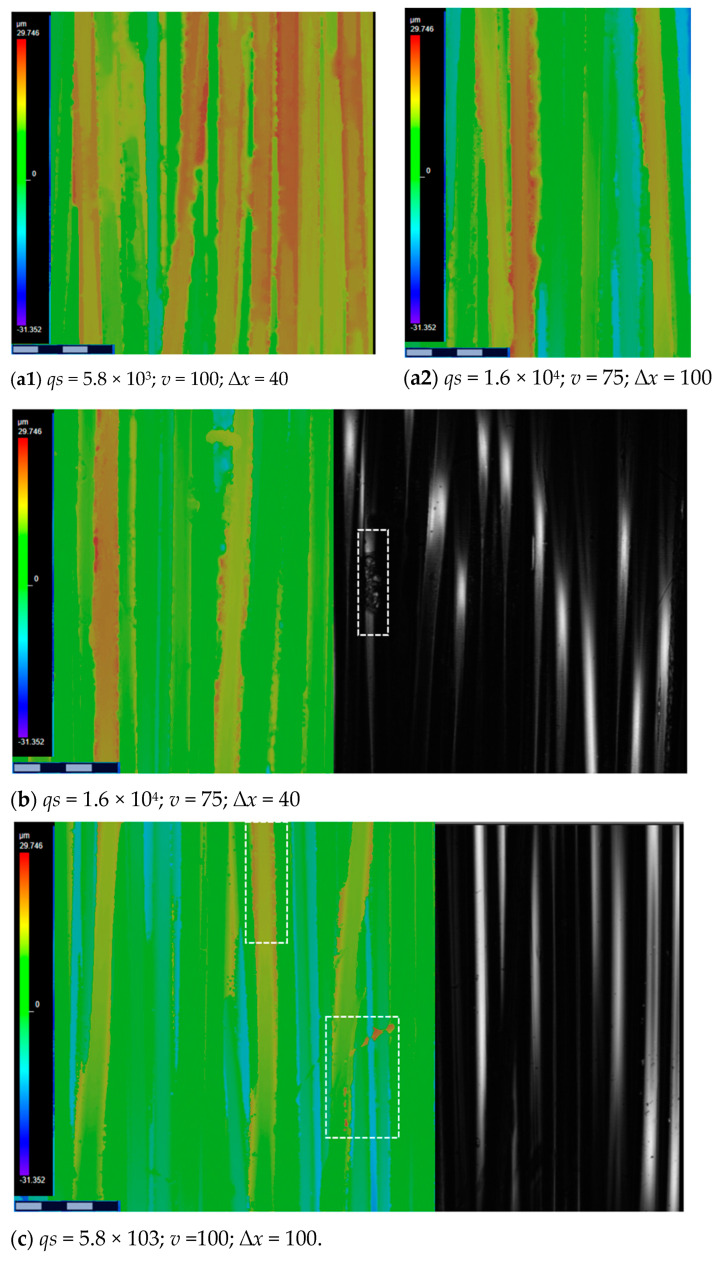
Selected micrographs of laser processed samples. (**a1**) 5.8 J/cm^2^, (**a_2_**) 21.3 J/cm^2^, (**b**) 21.3 J/cm^2^, (**c**) 5.8 J/cm^2^.

**Figure 14 polymers-17-02931-f014:**
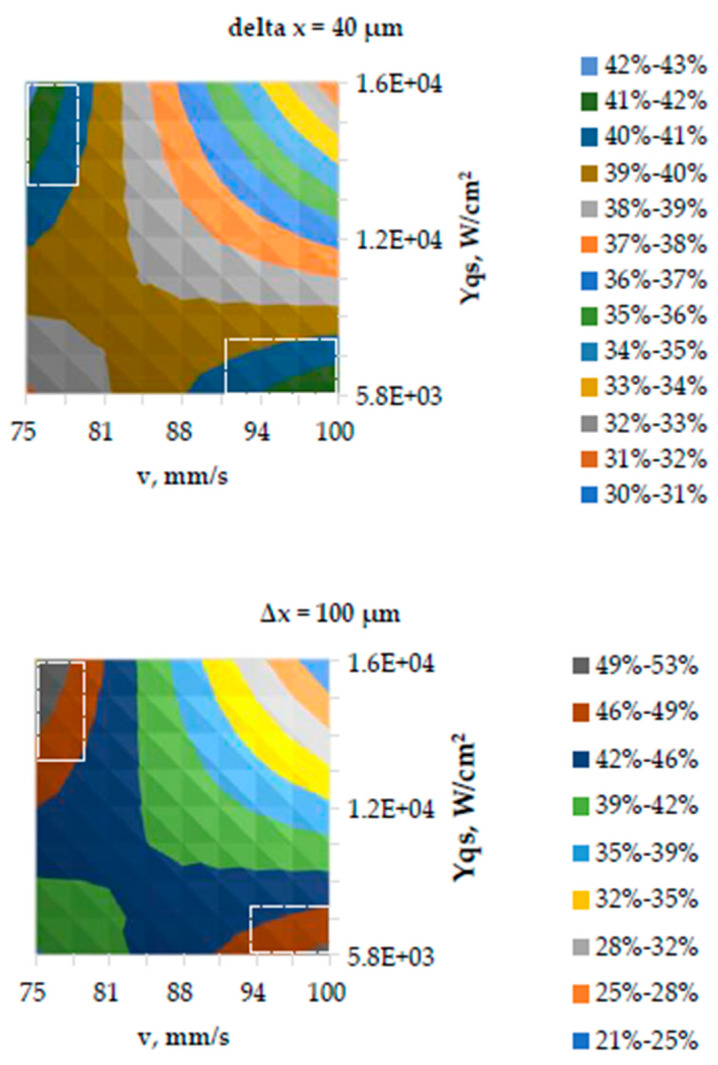
Response surface contour plot of surface roughness increases in relation to the raw sample.

**Figure 15 polymers-17-02931-f015:**
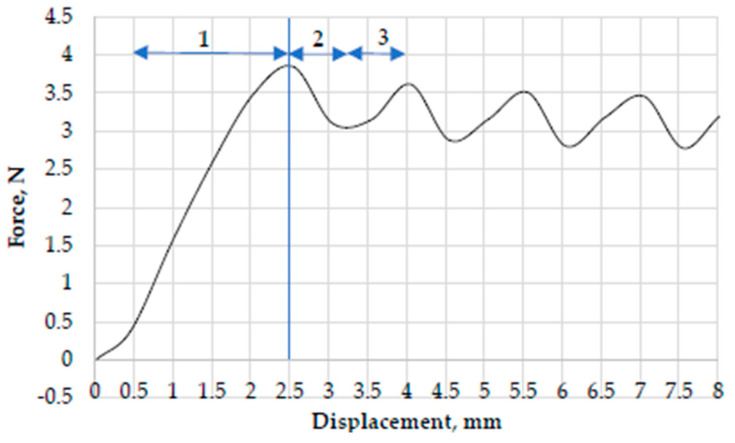
Crimp extension (1), Stick-slip (2), Accumulative retraction (3).

**Figure 16 polymers-17-02931-f016:**
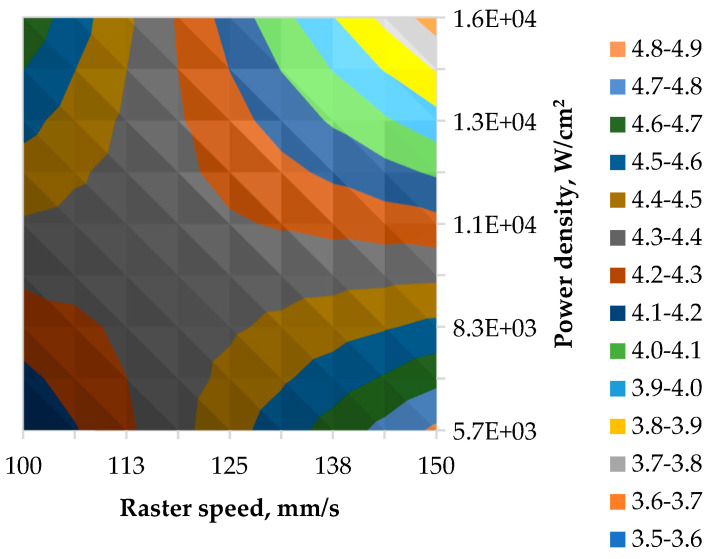
Contour plot of mean junction rupture force.

**Figure 17 polymers-17-02931-f017:**
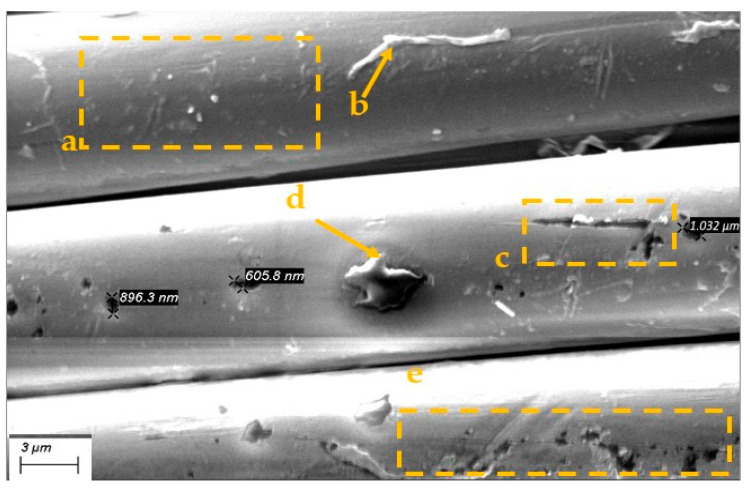
Laser processed self-organized hybrid micro- nanostructures KM2+ fibres surfaces. Mag 3000×. (**a**) nanoscale bumps, (**b**) locally torn fragment of fibers’ skin layer, (**c**) long groove combined with irregular form cavity, (**d**) nano- and micropores combination on a fiber surface with a deformed cross-section, (**e**) irregular shape microscale structure.

**Figure 18 polymers-17-02931-f018:**
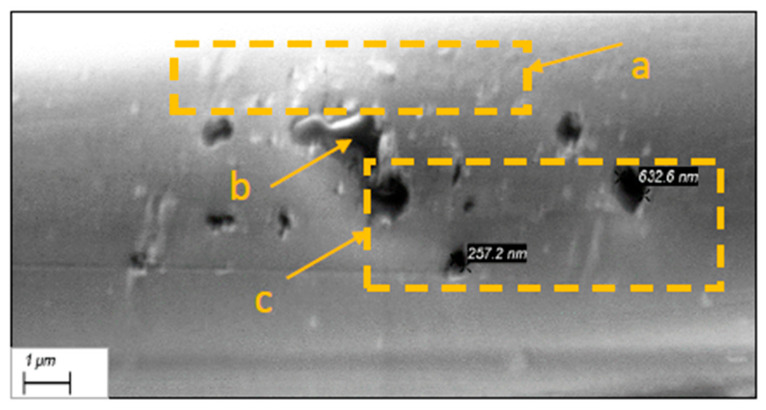
High resolution image of laser processed fibre surface. Mag 7000×. (**a**) dominate nanoscale bumps, (**b**) microscale protrusion, (**c**) submicron pores on the background of negligible sized nano protrusions.

**Table 1 polymers-17-02931-t001:** Single Fibre Tensile Testing Quantifications [51].

Kevlar^®^ Fibre Type	Elastic Modulus (GPa)	Tensile Strength (GPa)	Ultimate Strain (mm/mm)
K119	66.5 + 8.6	3.40 + 0.49	0.043 + 0.003
K29	72.8 + 6.9	3.30 + 0.39	0.042 + 0.002
** KM2 +	84.3 + 8.1	2.99 + 0.33	0.03 + 0.004
* K49	93.0 + 11.6	2.74 + 0.35	0.028 + 0.003

* K49 harsh post-processing heat treatments under tension at a high temperature. ** KM2+ intermediate intensity post-processing.

**Table 3 polymers-17-02931-t003:** The 2^2^ factorial design levels.

		−1	0	1	Interval
**Power density (W/cm^2^)**	*q_s_*	5.7 × 10^3^	1.1 × 10^4^	1.6 × 10^4^	5.1 × 10^3^
**Raster speed, mm/s**	*V*	1.0 × 10^2^	1.25 × 10^2^	15 × 10^2^	25
**Raster step between lines,** **µm**	*Δx*	80			

## Data Availability

The original contributions presented in this study are included in the article. Further inquiries can be directed to the corresponding author.

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
