# Peer review of "The Impact of CO2 Laser Treatment on Kevlar® KM2+ Fibres Fabric Surface Morphology and Yarn Pull-Out Resistance"

_polymers, 2025, doi:10.3390/polym17212931_

Round 1

Reviewer 1 Report

Comments and Suggestions for Authors

In this manuscript, in order to address the issue of the poor adhesion performance of Kevlar fibers, the author attempt to increase the surface roughness and resistance to the yarn pullout from the fabric without destroying the unique structure of the of Kevlar® KM2+fibres by adjusting the continuous wave (CW) CO2 laser parameters. The result was found that improvement of interface roughness contributes to the enhancement of yarn pull-out max strength and the laser processing enhances these mechanical properties and increases surface roughness without disrupting the underlying fibre architecture.

  1. The description of the laser processing process is not detailed enough. If possible, please add information such as the irradiation intensity, irradiation time and irradiation distance.
  2. If possible, please also provide information on the influence of laser treatment on the performance of samples with different molecular weights of Kevlar.
  3. If possible, please also provide the contact angle tests before and after the laser treatment.

Reviewer 2 Report

Comments and Suggestions for Authors

In its actual form the manuscript needs a twofold care review: on one hand the Introduction Section that extends on a third part of the manuscript should be reduced by focusing just in the aim and scope of the work; on the other side, the overall drafting between the different sections needs to be homogeneized.

About the scientific/technical content, it seems too much specific and somewhat empirical in spite of the image techniques used. From this perspective this reviewer has serious doubts about the suitability of this work to be published in a broad spectrum Journal as Polymers.

Comments on the Quality of English Language

The overall drafting between the different sections needs to be homogeneized.

Reviewer 3 Report

Comments and Suggestions for Authors

Paper's objective is to optimize continuous wave CO2 laser parameters to increase surface roughness, which they hypothesize will enhance yarn pull-out resistance without damaging the underlying fibers. While the premise is of interest to the field, the manuscript suffers from flaws. The work, as presented, at its current form, does not meet the standards for scientific publication. The core issues are mentioned below:

1. The abstract's claim of "…improving the adhesion of functional coatings…" is speculative. The manuscript contains no experiments or data related to the application or adhesion of any coatings. The abstract's claim of modifying the surface "…without destroying the unique structure of the of Kevlar KM2+ Fibers" is directly contradicted by (the authors' own data) in Figures 12 and 13, which are described as showing "unacceptable fiber degradation." The authors should either a)remove the claims from the abstract or re write them so they only describe what you actually measured.  Or b)add a small set of coating adhesion experiments (such as, peel or pull off tests on laser treated samples), show the results, and then revise the abstract to match those findings.

2. The current Discussion section cannot mechanistically link the laser induced surface features back to this structure. The analysis are superficial. I am expecting, in the discussion, explicitly link each surface change (grooves, pits, melt reflow) to a specific structural element of Kevlar like crystallites, lamellae, etc and cite relevant literature or trim the introduction to only the structural points you actually use, and add a short subsection that maps laser parameters to the micro structural response.

3. The text describes results in Figures 12 and 13 as "unacceptable fiber degradation" but includes these data points in the subsequent optimization analysis. An "optimization" study that includes known failure points in its dataset is logically invalid so either exclude the unacceptable data points from the regression and redo the optimization using only the valid region or keep the full dataset but add a "damage" flag, run a separate analysis that distinguishes acceptable from unacceptable outcomes, and discuss how the inclusion of the "bad" points changes the optimum.

4. More on that, the authors correctly identify in the introduction that "…it is necessary to protect the fiber from the massive ablation and considerable thermal damage…" However, they then proceed to include experimental outcomes that violate this (show "unacceptable damage") within their optimization model, invalidating the optimization. Maybe these two option can work here: a) add a constraint to the optimization that automatically discards any parameter set that causes the damage you identified. or b) first map the safe operating window with a focused set of experiments, then limit the optimization to that window only.

5. A sample size of N = 2 ("Two samples have been prepared for each variant…") is statistically indefensible. It is insufficient to estimate variance, conduct valid statistical tests, or build a reliable regression model. In my view this invalidates statistical claims in the paper. I recommend that you increase the number of replicates (at least five per condition) and redo all statistical analyses.

6. The regression equation (Equation 3) and corresponding response surface plots (Figure 15) are presented without any statistical validation metrics. Add the missing statistical metrics (R2, adjusted R2, ANOVA tables, pvalues, residual plots) directly to the manuscript or SI.

7. Insert a short, stand alone "Conclusion" section that restates the main findings, notes the limitations of this study, and suggests next steps. If you prefer not to add a separate section, add a "Conclusions" subsection at the end of the discussion that serves the same purpose.

8. The concluding claims of identifying 'optimal combinations' and achieving the 'greatest resistance increase' are baseless. Tone down the language (use "promising" or "potentially advantageous") and acknowledge the uncertainty. If you decided to Re‑run the analysis, perform without the compromised data, provide proper statistical validation, and only then make strong claims if they are truly justified.

9. The methodology for the "Quasi Static Yarn Pull Out Setup" omits the crosshead speed or pull out rate. To my knowledge yarn pull out force is rate dependent, making the reported JRF values meaningless. State the exact pull out speed you used (including units) and explain why you chose it.

10. The manuscript (including the SEM part and every other characterization) does not give the essential details, how the specimens were prepared, the exact test conditions, any coatings applied, the assumptions/calc made, and the way the raw data were processed and rendered. Please add all of these details for every characterization in the paper.

11. The confocal microscopy part doesn't say anything about how you treated the raw data before you got the roughness numbers,  no mention of leveling the images, filtering out noise, or any other processing steps. Please add those details. (see #10)

12. The properties of aramid fibers and laser material interactions are sensitive to test conditions, and their omission makes the experiments not reproducible. Add a table with the temperature and relative humidity for each experimental block (or the typical lab conditions) and comment on their possible influence.

13. There are two distinct figures both labeled as "Figure 16." 

14. The abstract claims a "50 - 53%" roughness increase, but the legend in Figure 15 shows the highest range is "53 - 56%". Was it because it was a very small region at the top left corner? reconcile the discrepancy, while providing a brief explanation for any regional variations in the response surface.

15. I strongly suggest that authors perform additional experiments that directly correlate measured roughness with pull out force across a broader set of laser conditions (including statistical analysis or simple simulations if possible). If not possible then despite decreasing the impact of the work but keep the claim as a hypothesis, clearly state that it is preliminary, and outline a concrete plan for future work (see #7) to test it (systematic parametric studies, micro mechanical testing).

Round 2

Reviewer 2 Report

Comments and Suggestions for Authors

Modifications made by authors improve nothing the first opinion of this reviewer. So the rejection of the manuscript remains

About the cover letter'content for v.2 manuscript: Congratulations to authors for their previous published work, but it is not that is under review process now, neither the: "on the topic we started."

Author Response

Thank you very much for the decision!  We agree with your comments.

1. Pag. 5 line 164. Is there any difference if the laser irradiates the material in vacuum or in air? Please specify the environment in the sentence where you refer to oxidation.  The term oxidation has been replaced with a more precise decomposition in relation to the text.
Explanations, as possible based on what is known so far, are added in Section 4, lines 637-653 and Conclusions, lines 704-706. 

2. Pag. 5 line 185 . What the STF stands for? Added in text: shear thickening fluid STF

3. Pag. 8 line 303. Please use subscript notation for numerical indices where appropriate. Changes made in the text

4. Pag. 8. Did the laser processing occur in vacuum? A schematic illustration of the laser processing setup would greatly enhance the clarity of the manuscript. Laser processing occurs in air.
Figure 5. Experimental setup of laser processing is added in the methodological section

5. Pag. 11 Fig. 6 The scale bar is missed. Changes made

6. Pag. 12 Fig. 7. Improve the quality of the Figure 7. Changes made.

7. In Fig 8 there are two images which should be distinguished and indicated in the label. Changes made.

Reviewer 3 Report

Comments and Suggestions for Authors

The manuscript improved significantly, good job! Some work required for authors to cover the remaining shortcomings of the manuscript.

  1. Please plot the measured roughness increase (Yqs or Rq) versus the corresponding JRF for every valid laser‑parameter set. Compute and report the correlation coefficient together with its statistical significance (p‑value). Discuss any outliers and what they reveal about the relationship.
  2. Kindly provide a full statistical data summary for readers in SI
  3. State explicitly the total number of measurements per experimental condition (like n = 20: two independent replicates, each comprising ten repeated roughness scans”). Present each result as mean ±â€¯standard deviation (or standard error) in tables/figures if havent.
  4. Add a concise methods paragraph (or supplementary information) that details:

   - image leveling/plane‑fitting, any noise‑reduction filtering  ?

   - outlier detection/removal criteria if any!

   - roughness metric (Rq) and any averaging procedure.

  1. Specify the exact thresholds that separate acceptable from unacceptable laser‑induced damage. Explain please, how these criteria were applied to filter the dataset before regression analysis.
  2. The tone and language shall be in accordance to the data . Re‑phrase statements to reflect the exploratory nature of the work. Its important that all claims are qualified by the experimental scope and acknowledged limitations. (Verbs: Prove vs. Suggest)

Author Response

Thank you very much for the comprehensive review, which allowed make appropriate corrections

1. Please plot the measured roughness increase (Yqs or Rq) versus the corresponding JRF for every valid laser‑parameter set. Compute and report the correlation coefficient together with its statistical significance (p‑value). Discuss any outliers and what they reveal about the relationship. 

Rq and JRF primary measurements were obtained in experiments of I and II step, respectively. Each of them is planned by applying the information obtained in step I to the planning of step II. As a result, both the ranges of laser processing parameter variations and the volumes of the obtained data series differ significantly, which does not provide the prerequisites for data collection for correlation analysis. In future studies, we will plan a correlation analysis.

2. Kindly provide a full statistical data summary for readers in SI. Made changes: statistical data units aligned with the SI system in text.

3. State explicitly the total number of measurements per experimental condition (like n = 20: two independent replicates, each comprising ten repeated roughness scans”). Present each result as mean ±â€¯standard deviation (or standard error) in tables/figures if havent. Made changes: added text “in total 20 primary measurements for each set of factors”. “Mean ±â€¯standard error” in Table 4 is added to those indicators that are important. Since the main one analyzed is JRF and its corresponding relative standard error does not exceed 1.7%, the error of the mean JRF increase was not calculated due to the low  â€¯standard error of the input information. Explanatory text added below table 4.

4. Add a concise methods paragraph (or supplementary information) that details: - image leveling/plane‑fitting, any noise‑reduction filtering?  - outlier detection/removal criteria if any! Primary measurements of samples roughness were taken before and after laser treatment to be able to evaluate the roughness changes separately for each sample and then calculate the average increase for every laser parameter set. Image leveling/plane‑fitting, noise‑reduction filtering not carried out given that most images were obtained with high-resolution microscopes by changing magnifications to observe increasingly smaller structures on the fiber surface. 

5. Specify the exact thresholds that separate acceptable from unacceptable laser‑induced damage. Explain please, how these criteria were applied to filter the dataset before regression analysis. Made changes: Text added “Laser‑induced damages controlled visually, by color and confocal images of confocal microscopy. Partial (the effect of carbonization is assessed based on changes in the color of the fibres surface) or complete decomposition (the disappearance of grouped fragments of the upper fibres) are classified as unacceptable changes”.